# Parental Pre and Postnatal Depression: The Longitudinal Associations with Child Negative Affectivity and Dysfunctional Mother–Child Feeding Interactions

**DOI:** 10.3390/children10030565

**Published:** 2023-03-16

**Authors:** Loredana Lucarelli, Laura Vismara, Irene Chatoor, Cristina Sechi

**Affiliations:** 1Department of Pedagogy, Psychology, Philosophy, University of Cagliari, 09123 Cagliari, Italy; 2Department of Psychiatry and Behavioral Sciences, The George Washington University, Washington, DC 20052, USA; 3Children’s National, Washington, DC 20010, USA

**Keywords:** prenatal parental depression, postnatal parental depression, child negative affectivity, feeding interactions

## Abstract

Background: Many studies have shown the influence of maternal perinatal depression on a child’s emotional and behavioral regulation ability; yet there is scarce research on the impact of the father’s perinatal depression on the caregiver–infant relationship and the child’s development. Methods: Through a longitudinal study, we investigated maternal and paternal depression and its association with infants’ emotionality and mother–infant feeding interactions The sample was constituted of 136 first-time parents (68 couples, and their full-term babies at 3 and 6 months old). At T1 (28th week of pregnancy), T2 (three months old), and T3 (at six months age) parents responded to the Edinburgh Postpartum Depression Scale. At Times 2 and 3, mothers and fathers completed the Infant Behavior Questionnaire, and recorded mother–infant interactions were coded by means of the Feeding Scale. Results: Statistical analyses indicated stability of maternal and paternal depression over time. Correlations emerged between mother’s higher depression scores, negative affective state during interactions at three months age, infant food refusal and mother–infant interactional conflict at six months age. Paternal higher depressive scores were associated with the mother–child interactional conflict. To finish, higher parental depression scores were related with infant negative emotionality. Conclusion: The current study confirms the relevance of embracing a cumulative risk model to support the child’s development with early caregiver-child interventions.

## 1. Introduction

Scientific literature has widely recognized that maternal depression has important short-, medium-, and long-term implications for their offspring in terms of emotional and behavioral dysregulation, cognitive delays, and physical health emerging from early negative, impaired caregiver-child interactions [1,2,3]. Similar influence has been shown both in mothers and fathers with depressive symptoms [4,5,6]. Indeed, fathers with perinatal depression report lower sense of parenting efficacy, higher parental distress [7] and lower involvement [8]. Within such a research framework, it is pivotal to analyze, as early as possible, the mutual influence between the infant and his/her environment and the interaction among the variables that take part in the child’s development, trying to recognize the risk factors significantly associated with impaired developmental trajectories.

A number of domains appear to be jeopardized by parental perinatal depression, including, in particular, feeding interactions [9,10,11,12,13]. These studies suggest that the observation of the regulatory processes within early infant-caregiver exchanges, such as mother–child feeding interactions, may provide crucial information to understand the complexity of a child’s development, better than studying maternal psychopathology alone. Moreover, maternal depression has been thoroughly studied among infants and children with feeding problems [14,15,16]. These mothers are either over controlling, disengaged, or inconsistent; showing a lack in responsiveness due to either their anxiety or anger in the context of feeding [17,18,19,20,21,22].

In relatively recent times, more attention has been drawn to the understanding of fathers’ impact upon their children’s development. Certainly, the massive entry of women into the world of work has led to an increase in fathers’ childcare responsibilities [23,24], that goes along with a broader cultural shift in Western societies as regards the role of fathers [24,25]. Overall, the studies that have been conducted to date seem to demonstrate that paternal parenting is correlated with the quality of his relationship with his offspring and their development; still, more studies are necessary to confirm and deepen these findings [24,26,27].

Spare studies address the role of fathers in the context of feeding interactions, confirming fathers’ lower sensitivity and higher intrusiveness in the presence of father–child interactional conflict and child’s feeding disorders [18,28]. In addition, paternal depression and parenting stress has been associated with children’s feeding problems [29,30]. Certainly, the quality of feeding interactions may be influenced by several variables attributable to either the parent, the child or their milieu [31,32,33].

With respect to the infant’s temperament, several studies have indicated that the more the child is impulsive, intense, inflexible the more likely that he will encounter early feeding problems. Such findings seem to be confirmed by mothers’ evaluation of their child’s regulatory abilities [34,35,36].

Of particular interest for feeding disorders is the temperamental factor of negative affectivity (NA), that has been “defined by scales of Sadness, Discomfort, Anger/Frustration, Fear, and loading negatively, Falling Reactivity/Soothability” [37] (p. 4), inevitably impacting the biological functions and emotional and behavioral response to internal and external stimuli.

Although the literature has focused primarily on maternal predictors of infant NA [38], it has been demonstrated that fathers’ depression and anxiety also seem to be associated with their child’s NA [39,40].

Considering the above, we apply a transactional perspective [41] to enhance the understanding of early feeding dysregulation. To our knowledge no study has longitudinally examined the associations among all the aforementioned variables; for such reasons, the current study intends to investigate the depressive risk occurring in mothers and fathers during pregnancy and in the postnatal period, as well as study the longitudinal association between perinatal parental depression levels and the child’s subsequent NA, and the quality of feeding mother–child interactions at three and six months of the baby’s age.

Specifically, this longitudinal study proposed three different time points: T1, the third trimester of pregnancy; T2, three months after a child’s birth; and T3, six months after a child’s birth. We recruited a sample of couple of parents where mothers and fathers showed prenatal depressive symptoms (Group A) and we compared them to a sample of mothers and fathers without symptoms, matched by the same sociodemographic characteristics (Group B). Thus, the purpose of this study was to:(1)assess mothers’ and fathers’ perceptions of their infant’s NA as well as the quality of mother–infant interactions during feeding to verify potential differences between the two groups (Group A vs. Group B);(2)explore changes in mothers’ and fathers’ perceptions of their infant’s NA and the quality of mother–infant interactions over time.

## 2. Study Design

The sample of the current study was drawn from a longitudinal study which started in February 2013 and ended in February 2016 aimed at investigating the relations between maternal and paternal depression and the development of their children’s regulatory capacity.

In the current study, we present data concerning parents who completed the first (T1), second (T2), and third point (T3) of the assessment at the third trimester of pregnancy, third, and sixth month after the child’s birth (Appendix A).

Figure 1 shows the study design and the measures at each time point (Appendix A).

## 3. Materials and Methods

### 3.1. Procedure

The study protocol was analyzed and approved by the Department of Psychology of the University of Cagliari ethics committee (N° 2/2013) on 5 February 2013. Expectant couples were recruited through snowball sampling from a pool of families attending prenatal courses in a large city in central Italy. Participation was voluntary, and participants had to meet the following inclusion criteria: (a) at least 18 years of age; (b) first pregnancy, (c) spontaneous pregnancy, and (d) no gestational pathology. Parents were excluded if they (a) were under treatment for depression or other psychopathological problems, or (b) were currently receiving psychological help. Eligibility depended on maternal and paternal adherence to the inclusion and exclusion criteria.

Participants gave their permission to provide data during three separate laboratory visits. First, at T1, mothers and fathers came to our laboratory and signed a written informed consent form and independently completed a questionnaire to evaluate perinatal depression. At T2 and T3, mothers and fathers were asked again to fill out the questionnaire to evaluate perinatal depression. At T2 and at T3 mothers and fathers completed a parent-report measure of their child’s temperament. In addition, at T2 and T3 mothers were asked to return with their babies to our laboratory to participate in a mother–child feeding interaction session.

### 3.2. Participants

Out of 162 participants (81 couples) who had initially entered the study (T1), 11% (9 couples) dropped out at T2 and 5.5% (4 couples) at T3. No difference emerged between the couples who dropped out of the study and those who were evaluated at all three time points in relation to sociodemographic characteristics or outcome measures.

The final sample was composed of 136 parents (68 couples) and their healthy babies (56% boys, 44% girls). The mothers’ age ranged from 20 to 44 years (M_Age_ = 34.81 years, SD = 4.66 years). The fathers’ age ranged from 20 to 51 years (M_Age_ = 37.68 years, SD = 6.05 years). As regards parental educational level, 5.8% of the mothers and 7.2 % of the fathers had an elementary school degree (at least 5 years within the education system), 37.7% of the mothers and 52.2% of the fathers had an upper secondary school degree (at least 12–13 years within the education system), and 56.5% of the mothers and 40.6% of the fathers had an undergraduate degree (at least 17 years within the education system).

Of the couples, 65.2% were married (*n* = 45 couples) and 58% of the mothers and 85.6% of the fathers were employed. Seventy-seven percent of parents (*n* = 53 couples) reported that the pregnancy was planned and only 5.8% revealed that the pregnancy was unexpected (*n* = 4 couples).

### 3.3. Measures

#### 3.3.1. Parental Depression (Exposure)

At T1, T2, and T3, mothers, and fathers completed the Edinburgh Postnatal Depression Scale (EPDS; [42,43]). This is a 10-item self-report questionnaire about mood experienced over the previous seven days.

The overall score is computed by adding items on a 4-point Likert scale (from 0 to 3), yielding a total range of 0–30. In the community screenings, the best cutoff to identify Italian people with a diagnosis of a severe or moderate major depression episode is 8/9, with a 94.4% sensitivity, 87.4% specificity and 58.6% positive predictive value [43].

In the current study, the internal consistency coefficient for the mothers was α = 0.80 at Time 1, α = 0.82 at Time 2, and α = 0.80 at Time 3; for the fathers, it was α = 0.77 at Time 1, α = 0.75 at Time 2, and α = 0.74 at Time 3.

#### 3.3.2. Infant Negative Affectivity (Outcome)

At both T2 and T3, mothers and fathers completed the Infant Behavior Questionnaire-Revised (IBQ-R; [37,44]). This is a 191-item parent-report form about temperament created to assess infants between ages 3 and 12 months. Both parents independently evaluated the frequency of their child behaviors on a 7-point scale (from “never” to “alway”). The IBQ-R yields 14 scales that form three higher-order factors: a positive affectivity/surgency factor; orienting/regulatory capacity factor, and a negative affectivity factor (NA). For the purposes of the current investigation, the broad NA factor was used. Negative affectivity consists of the following scales: sadness (“*Did the baby seem sad when the caregiver was gone for an unusually long period of time?*”), distressed reaction to limitations (“*When placed on his/her back, how often did the baby fuss or protest?*”), fear (“*How often during the last week was the baby startled by a sudden or loud noise?*”), and falling reactivity/rate of recovery from distress (“*When frustrated with something, how often did the baby calm down within 5 min?*”).

In the current study, the alpha coefficient(s) for NA factor score was α = 0.77 (T2) and α = 0.75 (T3), for mothers; and for fathers, α = 0.73 (T2) and α = 0.75 (Time 3).

#### 3.3.3. Mother–Child Feeding Interactions (Outcome)

At T2 and T3, we observed the mother-baby feeding interaction. We videotaped and coded the early dyadic feeding interactions by using the Italian version of the Observational Scale for Mother–Infant Interaction during feeding (SVIA) [19], the Italian validation of Chatoor’s feeding scale [45]; it allows to assess the quality of the mealtime exchanges according to four main subscales: affective state of the dyad, affective state of the mother, interactional conflict, and food refusal behavior. The feeding scale consists of 40 items evaluating several interactive behaviors that allow to identify normal and/or at-risk feeding relational dynamics between mothers and their 1-month–3-year-old child. Mother–child interactions were videotaped during a 20-min feeding session. Each item was scored on a Likert scale of 0 (none), 1 (a little), 2 (pretty much), and 3 (very much); then a global score was rated for each subscale.

The affective state (AS) of the mother subscale assesses the difficulty the mother has in expressing positive affect and the frequency and quality of negative affect. It also refers to the mother’s responsiveness to the child’s needs and capacity to favor mutual and sensitive exchanges. The higher the subscale score, the greater the difficulties shown by the mother in showing positive feelings and in understanding and tuning to the communicative signals of the child.

The interactional conflict (IC) subscale evaluates both the presence and intensity of the dyadic conflictual exchanges. The global subscale score is high when the mother forces the child to eat, when she is not flexible in regulating pauses and turn-taking and directs the meal according to her own emotions and intentions, rather than following the communicative cues given by the child who shows signs of distress and avoidance as a response to the mother’s intrusiveness.

The food refusal behavior subscale (FRB) evaluates the feeding patterns of the child, indicating food avoidance, insufficient food intake, and poor state regulation as emerging from negative affectivity. This subscale also examines noncontingent maternal behaviors during feeding interactions. A high score indicates a lack of dyadic reciprocal adaptation and a high frequency of food refusal.

Lastly, the affective state of the dyad subscale (ASD) evaluates the quality of affect within the mother–child relationship. A high score indicates a negative dyadic involvement featured by anger and hostility with the caregiver focusing on control of the child’s behavior, prone to limit his/her autonomy, which in turn leads the child toward distress.

Psychometric studies have confirmed satisfactory inter-rater reliability, construct, and discriminant validity [17,19]. The inter-rater reliability for the Italian version has been tested through the use of intraclass correlation coefficients, ranging from r = 0.82, *p* = 0.01, to r = 0.92, *p* = 0.01. The inter-rater reliability observed in the current study ranged between 0.79 and 0.89 (M = 0.83).

#### 3.3.4. Data Analysis

Due to the longitudinal study design, there were missing data; therefore, multiple imputation (MI) was used. Prior to performance of MI, the data were examined to confirm that missing values were missing at random (MAR). Then, the amount of missing data was evaluated to guarantee that less than 10% of data were missing across scale scores. The assumption of MAR was met, and the amount of missing data across scales (5–7%) was acceptable. Twenty datasets were imputed.

Descriptive statistics were computed for the exposure (groups—depressed and non- depressed parents and depressive symptoms—EPDS scores) and outcomes (infant NA and mother–child feeding interactions), as well as for potential confounder variables (parental educational level, marital status, and baby sex).

We calculated percentages for categorical data and descriptive statistics (means, standard deviations), and normality statistics for continuous data. Bivariate correlations between potential confounders with exposure and outcome variables were examined.

To examine if there were changes in parents with depressive symptomatology based on EPDS cutoff scores, the Friedman test was used. Repeated measures ANOVAs were conducted on EPDS mean scores to examine differences across time points, using the Greenhouse–Geisser adjustment.

We used Pearson’s r to examine correlations among the mothers’ and fathers’ depressive symptoms, the perceived infants’ NA score and the quality of mother–infant interactions during feeding.

Analyses of group differences (depressed vs. nondepressed groups) in child NA and in the quality of mother–infant interactions during feeding across time (T2 and T3) were examined using a mixed-design ANOVA.

To evaluate the significant effects, the Bonferroni correction for multiple comparisons was applied. Effect sizes were estimated using partial eta squared (η_p_^2^).

We used G*Power analysis software to determine the number of participants necessary. Based on repeated measures ANOVA, the minimum number of participants required was determined to be 44 for an effect size of 0.25, a statistical power of 0.85, and a significance level of 0.05, using two groups, and three measurements.

## 4. Results

### 4.1. Descriptive Analyses

None of the potential confounder variables were linked with the exposure and/or outcomes under analysis.

Results from all analyses were showed as unadjusted.

The rates of depression in the prenatal and postnatal period were higher in mothers than fathers.

Twenty-nine percent of the mothers scored higher than the EPDS proposed cutoff during T1.

At three months postpartum (T2), 31% of the mothers scored beyond the EPDS cut score and at six months after birth (T3) 24% of mothers scored beyond the EPDS cut score.

To evaluate the differences in medians for maternal depression across time, a Friedman test was conducted. The test was not significant, χ^2^ (3, *n* = 68) = 4.200, *p* = 0.122 showing a stability of maternal depression over time. Similarly, the repeated measures ANOVA showed no significant difference in EPDS mean scores F (2, 67) = 2.445, *p* = 0.095, ƞ^2^ = 0.035, indicating no differences across the three time points (Table 1).

Twenty-two percent of the fathers scored higher than the EPDS proposed cutoff during T1.

At three months postpartum (T2), 18% of the fathers scored beyond the EPDS cut score and at six months after birth (T3) 15% of fathers scored beyond the EPDS cut score.

To evaluate the differences in medians for paternal depression across time, a Friedman test was conducted. The test was not significant, χ^2^ (3, *n* = 68) = 1.181, *p* = 0.405 showing a stability of paternal depression over time. Similarly, the repeated measures ANOVA showed no significant difference in EPDS mean scores F (2, 67) = 0.045, *p* = 0.950, ƞ^2^ = 0.001, indicating no differences across the three time points (Table 1).

### 4.2. Correlations among Parental Depressive Symptoms, Infants’ NA, and Mother–Infant Feeding Interactions

Statistically significant correlations were found between mothers’ and fathers’ EPDS scores at all time points (r = 0.460, *p* < 0.001 at T1, r = 0.496, *p* < 0.001 at T2 and r = 0.459, *p* < 0.001 at T3). Table 2 presents the results of correlation analyses for mothers’ and fathers’ depressive symptoms (EPDS scores) and their perceptions of their infant’s negative affectivity (NA factor IBQ-R score) and mother–infant feeding interactions (SVIA subscale scores).

Mothers’ ratings of their infant’s NA at T2 and T3 were correlated with their EPDS scores at all three time points. Though weaker, fathers’ ratings of their infant’s negative affectivity (NA) at T2 and T3 were correlated with their EPDS scores at all three time points.

Significant correlations were found between prenatal and postnatal mothers’ depression scores and negative affective state during interactions and mother–child interactional conflict at three and six months of age. Also, significant correlations were found between prenatal and postnatal mothers’ depression scores and child’s food refusal at 6 months. Finally, fathers’ prenatal and postnatal depressive scores correlated with the mother–child interactional conflict at T2 and T3; fathers’ depressive scores at T3 also correlated with child’s food refusal at six months old.

### 4.3. Evolution and Differences between Groups in Infant Negative Affect from Three to Six Months

Mothers and fathers were divided into depressed and nondepressed groups based on their EPDS score during the third trimester of pregnancy (T1).

The demographic variables were similar for the depressed (Group A) and nondepressed (Group B) mothers as well as for the depressed and nondepressed fathers at T1.

A 2 (Group: A vs. B) × 2 (time points: T2 and T3) repeated measures ANOVA, separately for mothers and fathers, was performed for each NA scale and NA factor score. Means and standard deviations are shown in Table 3.

With regard to mothers, there were significant main effects of group for distress to limitations, fear, and sadness scale as well as for the NA factor scale. Depressed mothers were more likely to report higher sadness, fear, and NA factor scores than nondepressed mothers.

No significant main effect of time was obtained, therefore neither the NA scale scores nor the NA factor score changed significantly from T2 to T3 (Table 3).

Finally, we did not find significant interactions between group and time points for any NA scales or NA factor scale.

With regard to fathers, there were significant main effects of group for fear scale and NA factor scale. Depressed fathers were more likely to report higher fear and NA factor scores than nondepressed fathers.

No significant main effect of time was obtained, therefore, neither NA scale scores nor the NA factor score changed significantly from T2 to T3 (Table 3). Finally, we did not find significant interactions between group and time points for any NA scales or NA factor scale.

### 4.4. Evolution and Group Differences in Mother–Child Feeding Interactions from Three to Six Months

Mothers were divided into depressed and nondepressed groups based on their EPDS score during the third trimester of pregnancy (T1).

A 2 (Group: A vs. B) × 2 (time points: T2 and T3) repeated measures ANOVA was performed for each subscale score of SVIA. Means and standard deviations are shown in Table 4.

There were significant main effects of group for the AS of the mother, the IC, and for the FRB scales. The mother–child dyads of Group A were more likely to report a higher affective state of the mother, IC and FRB scores than the mother–child dyads of Group B.

A significant main effect of time was obtained, for interactional conflict F (1, 66) = 13.548, *p* = 0.001, ƞ^2^ = 0.175 and for food refusal behavior F (1, 66) = 31.278, *p* = 0.000, ƞ^2^ = 0.335 showing that the scores of these scales increased from T2 to T3 in both groups.

Finally, a significant Time × Group was obtained for the food refusal behavior F (1, 66) = 16.622, *p* = 0.000, ƞ^2^ = 0.211. This significant interaction was interpreted using a follow-up test that examined the effect separately at T2 and T3.

The results showed that only at T3, did the mother–child dyads of Group A report higher scores in FRB than the mother–child dyads of Group B (F (1, 66) = 10.207, *p* = 0.002, ƞ^2^ = 0.141).

## 5. Discussion

Maternal and paternal depressive symptoms are known to be related to impaired parenting behaviors and children’s emotional dysregulation [46,47,48,49,50]; following such observations, the current study investigated the association between prenatal and postnatal parental depression levels, their infants’ NA, and the quality of feeding mother–child interactions at three and six months’ old.

In terms of maternal and paternal depression, mothers showed higher depression scores than fathers, corroborating the finding that women are more susceptible to depression than men in the perinatal period [4,51,52]. Indeed, epidemiological studies indicate that about 25% of women postpartum [53,54,55] and 10% of fathers are at risk of depression [56,57] during the perinatal period. In addition, statistically significant correlations were longitudinally found between mothers’ and fathers’ depression scores. This result is in line with previous data that observed a frequent association between maternal and paternal depression during perinatality [58]. Although timing seems relevant in the impact of parental depression on the child’s outcome, our study shows no difference across the three time points. This may be due to the fact that we focused our longitudinal evaluation up to age six months, which seems to constitute a specific sensitive period in the association between parental depression and their offspring’s developmental sequalae [50,57].

Regarding the association between parental depression and child’s negative affectivity, our study showed that depressed mothers were more likely to report higher sadness, fear, and NA factor scores than nondepressed mothers, whereas depressed fathers were more likely to report higher fear and NA factor scores than nondepressed fathers. Several studies seem to confirm our findings: both maternal and paternal depression are predictive of infant NA [38,59]. It may be suggested that maternal and paternal depression decreases the caregiver’s sensitivity and ability to recognize their child’s mental state; as a consequence, the caregiver is not able to regulate the infant; thus, children may express higher levels of negative affectivity and reactivity. In turn, the infant’s fussiness, intense crying, and dysregulated patterns may increase the likelihood of depressive symptomatology in their parents, fueling a vicious cycle that may be revealed in either an intrusive/controlling or withdrawn/affectionless interaction.

Indeed, our study found a relation between prenatal and postnatal mothers’ depression scores and negative affective state during interactions and mother–child IC at three and six months. Also, significant correlations were found between prenatal and postnatal mothers’ depression scores and child food refusal at six months. Depressed mothers had more difficulty expressing positive feelings, empathically understanding, and responding to their child’s communicative cues compared to nondepressed mothers. Furthermore, depressed mothers forced the child to eat, were not flexible in regulating pauses and turn-taking, and directed the meal according to their own emotions and intentions rather than following the communicative feedback given by the child than nondepressed mothers. Finally, depressed mothers were less attuned and synchronized to the child’s signals and arbitrarily interrupted the meal, causing discomfort to the child than mothers who were not depressed. It is possible that infants of depressed mothers may display higher dysregulated behaviors and affectivity and lower social engagement [32,60] increasing the likelihood of need for control by the mother of the infant who responds by opposition during the feeding, thus increasing the levels of conflict between them [17]. In addition, our study shows that depressed mother–child dyads reported higher scores in food refusal behavior only at six months, when the child increases its autonomy and depressed mothers may feel the need to increase their control over the infant.

Interestingly, fathers’ prenatal and postnatal depressive scores correlated with the mother–child IC at three and six months of age, and fathers’ depressive scores at 6 months also correlated with the child’s food refusal. These results seem to confirm, on one hand, that fathers’ depression effects maternal styles of interaction [8,61,62], and on the other that, during the early stages of development, fathers play a significant, independent role in impacting their child’s regulatory processes, including feeding [1,23,63,64]. Furthermore, it is important to note that these results are in line with the more recent approach to the study of early feeding difficulties that recalls the need to consider the complexity of dynamics both at the dyadic and the triadic level of relationships, as emerging from the observation of the whole family system. This perspective allows the consideration of different factors which contribute independently to the development of the child’s socio-emotional functioning [28,63,65].

## 6. Limitations

The present study has some limitations. The sample size was rather small, with several implications for data analysis and results. In particular, a small sample size impedes the inclusion of more variables in the analysis, reducing the possibilities for testing complex models of relationships among the variables. In addition, our sample was composed of parents with medium/high education. Therefore, the results cannot be generalized to samples with lower education. Another limitation is that we did not evaluate the quality of father–child feeding interactions, nor the triadic feeding interactions that are yet our next objectives.

Although we used longitudinal data and applied accurate longitudinal methods to reduce potential biases from unmeasured child and parents’ characteristics, we are aware that the understanding of the complexity of child feeding difficulties cannot be limited to the considered individual and relational variables that we evaluated in the study; the inclusion of cultural variables, for instance, would add much information about the involved processes and, consequently, would increase the efficacy of diagnosis and interventions.

## 7. Conclusions

In conclusion, our study highlights the importance to screen both parents during pregnancy and postpartum for depressive symptomatology. Indeed, although today it is recognized and shared that family dynamics have a significant impact on the development and adaptation of the child, the mechanisms by which this occurs have been poorly analyzed; our contribution highlights the negative consequences that parental depression may have on children’s emotional and behavioral regulation, at the basis of all domains of individual and relational functioning [66].Thus, observing early feeding interactions may offer a valuable chance to create preventive interventions.

## Figures and Tables

**Figure 1 children-10-00565-f001:**
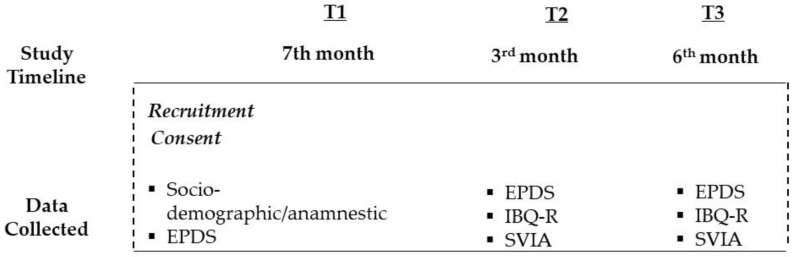
Study timeline and data collected at each of the three time points. Note: EPDS: Edinburgh Postnatal Depression Scale; IBQ-R: Infant Behavior Questionnaire-Revised SVIA: Observational Scale for Mother–Infant Interaction during Feeding.

**Table 1 children-10-00565-t001:** Distribution of EPDS unadjusted mean scores of mothers and fathers.

	T1 (Pregnancy)		T2 (3 Months)		T3 (6 Months)	
	Mothers	Fathers	Mothers	Fathers	Mothers	Fathers
EPDS						
Mean score (SD)	6.01 (3.93)	4.46 (3.53)	5.71 (3.19)	4.41 (3.62)	5.45 (2.77)	4.50 (2.82)
Depressed (≥8/9) N (%)	20 (29.4%)	15 (22%)	21(30.9%)	12 (17.6%)	16 (23.5%)	10 (14.7%)

Note: T1 = during the third trimester of pregnancy T2 = at 3 months after birth; T3 = at 6 months after birth; EPDS, Edinburgh Postnatal Depression Scale.

**Table 2 children-10-00565-t002:** Correlations between mothers’ and fathers’ EPDS unadjusted mean scores and IBQ-R scores and SVIA scores.

	EPDS Mothers T1	EPDS Mothers T2	EPDS Mothers T3	EPDS Fathers T1	EPDS Fathers T2	EPDS Fathers T3
NA Mothers T2	0.421 ***	0.378 **	0.374 **	0.330 **	0.369 **	0.230
NA Mothers T3	0.339 **	0.256 *	0.344 **	0.259 *	0.282 *	0.139
NA Fathers T2	0.142	0.206	0.188	0.315 **	0.277 *	0.244 *
NA Fathers T3	0.051	0.136	0.147	0.327 **	0.271 *	0.243 *
AS T2	0.316 **	0.338 **	0.394 **	0.081	0.080	0.066
IC T2	0.271 *	0.286*	0.299 *	0.402 ***	0.291 *	0.296 *
FRB T2	0.049	−0.009	−0.043	−0.060	−0.182	−0.052
ASD T2	−0.022	0.088	0.041	−0.028	−0.174	−0.062
AS T3	0.170	0.277 *	0.307 *	0.127	0.069	0.118
IC T3	0.349 **	0.315 **	0.259 *	0.380 **	0.281 *	0.246 *
FRB T3	0.292 *	0.313 *	0.245 *	0.231	0.210	0.263 *
ASD T3	0.169	0.156	0.168	0.228	0.053	0.152

Note: T1 = during the third trimester of pregnancy T2 = at 3 months after birth; T3 = at 6 months after birth; EPDS, Edinburgh Postnatal Depression Scale; NA = negative affectivity factor of the Infant Behavior Questionnaire-Revised (IBQ-R); AS = The affective state of the mother subscale of observational scale for mother–infant interaction during feeding (SVIA); IC = the interactional conflict subscale of SVIA; FRB = the food refusal behavior subscale of SVIA; ASD = the affective state of the dyad subscale of SVIA. * *p* < 0.05; ** *p* < 0.01; *** *p* < 0.001.

**Table 3 children-10-00565-t003:** Descriptive statistics and group comparisons of the IBQ-R unadjusted mean scores.

Mothers	T2		T3			
	GA (N = 20) M (SD)	GB (N = 48) M (SD)	GA (N = 20) M (SD)	GB (N = 48) M (SD)	F (1, 66)	η^2^
DIS	3.59 (0.88)	2.79 (1.07)	3.44 (0.79)	2.83 (1.15)	9.607 **	0.127
FEAR	2.55 (0.90)	1.58 (0.75)	2.57 (0.90)	1.95 (0.84)	22.962 ***	0.258
FAL	4.46 (1.17)	4.23 (1.87)	4.50 (1.97)	3.74 (1.97)	1.703	0.025
SAD	3.90 (0.79)	2.33 (1.28)	3.42 (1.24)	2.60 (1.27)	21.868 ***	0.249
NA	14.40 (2.34)	10.93 (3.569)	13.92 (2.32)	11.12 (4.12)	17.164 ***	0.206
	**T2**		**T3**			
**Fathers**	**GA (N = 15)** **M (SD)**	**GB (N = 53)** **M (SD)**	**GA (N = 15)** **M (SD)**	**GB (N = 53)** **M (SD)**	**F** **(1, 66)**	**η^2^**
DIS	3.47 (1.18)	2.61 (1.53)	2.98 (1.20)	2.35 (1.73)	3.587	0.052
FEAR	2.93 (1.14)	1.59 (1.00)	2.40 (1.29)	1.53 (0.7)	14.743 ***	0.183
FAL	3.99 (1.40)	3.75 (2.31)	3.85 (1.86)	3.19 (2.41)	0.708	0.011
SAD	2.96 (1.35)	2.20 (1.49)	2.73 (1.31)	2.16 (1.70)	3.242	0.047
NA	13.35 (4.13)	10.15 (5.77)	11.496 (5.58)	9.23 (6.65)	4.057 *	0.058

Note: GA: Participants with depressive symptoms at T1; GB: Participants without depressive symptom at T1; T2 = 3 months after birth; T3 = 6 months after birth; DIS = Distress to limitations scale of IBQ-R; FEAR = fear scale of IBQ-R; FAL = Falling reactivity/rate of recovery from distress scale of IBQ-R; SAD = sadness scale of IBQ-R; NA = negative affectivity factor of IBQ-R. * *p* < 0.05; ** *p* < 0.01; *** *p* < 0.001.

**Table 4 children-10-00565-t004:** Descriptive statistics and group comparisons of the SVIA unadjusted mean scores.

Mothers	T2		T3			
	GA (N = 20) M (SD)	GB (N = 48) M (SD)	GA (N = 20) M (SD)	GB (N = 48) M (SD)	F (1, 66)	η^2^
AS	22.65 (2.85)	20.46 (3.19)	22.15 (3.99)	20.29 (3.52)	5.306 *	0.074
IC	7.40 (2.91)	5.33 (3.85)	9.60 (3.05)	7.04 (3.70)	8.567 **	0.118
FRB	0.40 (0.94)	0.50 (0.79)	1.85 (1.90)	0.73 (0.92)	16.622 **	0.211
ASD	1.90 (1.90)	2.31 (2.03)	2.40 (1.53)	1.91 (1.60)	0.008	0.000

Note: GA: Participants with depressive symptom at T1; GB: Participants without depressive symptom at T1; T2 = at 3 months after birth; T3 = at 6 months after birth; AS = The affective state of the mother subscale of observational scale for mother–infant interaction during feeding (SVIA); IC = the interactional conflict subscale of SVIA; FRB = the food refusal behavior subscale of SVIA; ASD = the affective state of the dyad subscale of SVIA. * *p* < 0.05; ** *p* < 0.01.

## Data Availability

The data that support the findings of this study are available on request from the methodologist of the study CS. The data are not publicly available due to information that could compromise the privacy of research participants.

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
