# Peer review of "Parental Pre and Postnatal Depression: The Longitudinal Associations with Child Negative Affectivity and Dysfunctional Mother–Child Feeding Interactions"

_children, 2023, doi:10.3390/children10030565_

Round 1

Reviewer 1 Report

Thank you for giving me the opportunity to review this manuscript.

I think this manuscript is interesting.

It is still better to revise the manuscript.

1) Please describe the study design with a commonly used term in the title and the abstract. Was this a cross-sectional study?

2) Please attach the STROBE checklist and fill in the page numbers. Please describe the study design by using the PECO (participants, exposure, control and the outcomes) format.

3) Please clearly define all potential confounders. Please describe any efforts to address potential sourse of bias. Please explain how sample size was arrived at. Please describe all statistical methods to control for confoundings.

4) Please describe any characteristics of study participants and potential confounders in the result section. Please summarize the numbers of missing data at each stage.

5) Please describe unadjusted estimates and, if applicable, confounder-adjusted estimates and their precisions. Please make clear, if applicable, which confounders were adjusted and why they were included.

I think it is better to revise the manuscript.

Author Response

Responses to the reviewers

We would like to thank the Editor and the reviewers for their comments to improve the quality and clarity of our manuscript.

As required by the Editor, we tracked all changes, included the ones related to the reduction of our repeated rate.

Reviewer 1

Please describe the study design with a commonly used term in the title and the abstract. Was this a cross-sectional study?

Thank you for your proposal; we added the word longitudinal to both the title and the abstract.

Please attach the STROBE checklist and fill in the page numbers. Please describe the study design by using the PECO (participants, exposure, control and the outcomes) format.

Thank you for such request. We attach the STROBE Checklist as well as the PECO table. In addition, we added to exposure and outcome to the variable description to make clearer such characteristics.

Please clearly define all potential confounders. Please describe any efforts to address potential sourse of bias. Please explain how sample size was arrived at. Please describe all statistical methods to control for confoundings.

Thank you for this comment. We clarified on page 6 lines 251-257.

Please describe any characteristics of study participants and potential confounders in the result section.

Thank you. We added a comment on this request at page 6 lines 279-280.

 Please summarize the numbers of missing data at each stage.

Thank you; as regards missing data we confirm what already described at page 6 lines 254-250.

Please describe unadjusted estimates and, if applicable, confounder-adjusted estimates and their precisions. Please make clear, if applicable, which confounders were adjusted and why they were included.

Thank you; we added the required information.

I think it is better to revise the manuscript.

We hope that the revised version of the manuscripts is clearer.

Reviewer 2 Report

the current study investigated the association between prenatal and postnatal 324 parental depression levels, their infants’ negative affectivity and the quality of feeding 325 mother-child interactions at three and six  months of the baby.

I have several comments to improve the article:

In the literature review chapter, there is a lack of reference to the literature that talks about the father's role in the development of the baby and toddler, in the historical context. Why was the father not treated as a significant figure in research in the past, while in recent years he is becoming a significant figure? For this purpose, articles dealing with the transition to fatherhood and the involvement of fathers in raising children can be cited.

In the method chapter it is stated that the data were collected ten years ago. Is there a reason why no article has been written to date? Is it an article that is based on data that the authors have used in the past?

It is recommended to present a diagram of the research model so that it is easy to follow the variables measured at each point in time.

Are the relationships found between the fathers' variables and the mother's variables only related to the present or also belong to the match that existed between them even before the birth of the baby?

In the conclusions chapter it is important to refer to the theoretical contribution of the research and not only to the practical contribution.

Author Response

Responses to the reviewers

We would like to thank the Editor and the reviewers for their comments to improve the quality and clarity of our manuscript.

As required by the Editor, we tracked all changes, included the ones related to the reduction of our repeated rate.

 Reviewer 2

The current study investigated the association between prenatal and postnatal 324 parental depression levels, their infants’ negative affectivity and the quality of feeding 325 mother-child interactions at three and six  months of the baby.

I have several comments to improve the article:

In the literature review chapter, there is a lack of reference to the literature that talks about the father's role in the development of the baby and toddler, in the historical context. Why was the father not treated as a significant figure in research in the past, while in recent years he is becoming a significant figure? For this purpose, articles dealing with the transition to fatherhood and the involvement of fathers in raising children can be cited.

Thank you for your suggestion. We added to the introduction this relevant elements and related references (p.2 lines 63-70).

In the method chapter it is stated that the data were collected ten years ago. Is there a reason why no article has been written to date? Is it an article that is based on data that the authors have used in the past?

The manuscript is part of a larger longitudinal study. Indeed, we published several papers within the funded research; however, this manuscript is the first one to consider feeding interactions. Coding observations is time-consuming and highly specialized; therefore, it is among the last works within the recruited sample. We added a paragraph on the study design that we hope clarifies such issue.

It is recommended to present a diagram of the research model so that it is easy to follow the variables measured at each point in time.

Thank you; we added the suggested diagram (Figure 1).

Are the relationships found between the fathers' variables and the mother's variables only related to the present or also belong to the match that existed between them even before the birth of the baby?

The study is longitudinal; three time points are considered; thus, scores at pregnancy (T1) are evaluated and included in the analyses (page 7, lines 306-308).

In the conclusions chapter it is important to refer to the theoretical contribution of the research and not only to the practical contribution.

Thank you, we added theoretical contribution to the conclusions (lines 476-479), in addition to what was already stated in the discussion.

Reviewer 3 Report

Dear Authors

Very good effort on a fairly important topic. However, there are some issues that need to be clarified in my opinion in order for the article to be publishable.

Identify the study as a research study and specifically, a cross-sectional study.

Why did it take so long for the research to be submitted for publication?

The research approval number is nowhere to be seen, nor is the university identified.

“Out of 160 participants (81 couples)” has there been a mistake?

Put a citation in the psychometric tool EPDS, Mother-child feeding interactions, and Infant negative affectivity. Who were its creators, how did you get it? Translation needed...

To the limitations of the study, it should be added that confounding factors or additional variables were not examined.

Good luck! 

Author Response

Responses to the reviewers

We would like to thank the Editor and the reviewers for their comments to improve the quality and understanding of our manuscript.

Reviewer 3

Identify the study as a research study and specifically, a cross-sectional study.

Thank you; we further clarified that the study is longitudinal.

Why did it take so long for the research to be submitted for publication?

The manuscript is part of a larger longitudinal study. Indeed, we published several papers within the funded research; however, this manuscript is the first one to consider feeding interactions. Coding observations is time-consuming and highly specialized; therefore, it is among the last works within the recruited sample. We added a study design paragraph that hopefully clarifies this issue.

The research approval number is nowhere to be seen, nor is the university identified.

Thank you; we added the approval number in the procedure section.

“Out of 160 participants (81 couples)” has there been a mistake?

We are sorry for the typo: it is 162 couples, as now correctly reported.

Put a citation in the psychometric tool EPDS, Mother-child feeding interactions, and Infant negative affectivity. Who were its creators, how did you get it? Translation needed...

We report the references to both the original and the Italian validation of the applied instruments. References n.42, 43, 37, 44, 19 and we now added the reference of Chatoor (45), that we did not include previously because the Italian validation showed different factors.

To the limitations of the study, it should be added that confounding factors or additional variables were not examined.

Thank you, we added this relevant limit to the dedicated section.

Good luck! 

Thank you!

Round 2

Reviewer 1 Report

I think this manuscript would be suitable for publication in this journal.

Reviewer 3 Report

Dear Authors

Congratulations! 

You made a great job and you answered all my comments!

Yours Sincerely